# Potential Benefits of a ‘Trauma-Informed Care’ Approach to Improve the Assessment and Management of Dogs Presented with Anxiety Disorders

**DOI:** 10.3390/ani14030459

**Published:** 2024-01-31

**Authors:** Claire Lorraine Corridan, Susan E. Dawson, Siobhan Mullan

**Affiliations:** 1Animal Welfare & Ethics, UCD School of Veterinary Science, Belfield, D04 V1W8 Dublin, Ireland; siobhan.mullan@ucd.ie; 2Research Fellow in Psychology, University of Manchester, Manchester M13 9PL, UK

**Keywords:** canine welfare, canine fear aggression, post-traumatic stress, trauma-informed care and adverse early experience in puppies

## Abstract

**Simple Summary:**

Trauma-informed care (TIC) is an approach which has been utilised in human psychology for many years now. TIC considers how important early experience is in determining lifelong responses to challenging situations, how individuals respond to stress, how they overcome it, and their ability to develop and sustain resilience. There are a number of scientific publications which consider the importance of early experience in animals, both in utero and during their early development. This paper considers aspects of TIC approaches for humans which might be applied in dogs, focusing on both prevention of behavioural problems, by protecting puppies from adverse early experiences, and also, assessment of shelter dogs or those presented for problematic behaviours. A TIC approach for dogs would result in the following: the realisation that adverse early experience has significant consequences for canine welfare; recognising that where dogs respond in an apparently irrational or over the top manner, it may be the result of previous trauma; people involved in the care of these dogs must respond with empathy, understanding, and practical solutions to improve the welfare of the dog, while avoiding the need to re-traumatise them in as part of the diagnostic or treatment processes.

**Abstract:**

Dog caregiver reporting on the spectrum of fearful–aggressive behaviours often describes ‘unpredictable’ or ‘exaggerated’ responses to a situation/animal/person. A possible explanation for these behavioural responses considers that the dog is reacting to triggered memories for which the dog has a negative association. For many dogs undergoing veterinary behavioural treatment or rehabilitation through a canine rescue organisation, the assessing clinician relies on “proxy” reporting of the history/background by a caregiver (dog owner, foster carer, or shelter personnel). Detailed information on the event or circumstances resulting in this negative association may be limited or absent altogether. Consideration of a trauma-informed care (TIC) approach, currently applied in a wide range of human psychology and social care fields, may be helpful in guiding the clinical approach taken. The literature relating to adverse early experience (AEE) and trauma-informed care (TIC) in puppies/dogs compared to children/adults was evaluated to identify common themes and conclusions identified across both species. In the absence of known/identifiable trauma, behavioural assessment and management should consider that a ‘problem’ dog may behave as it does, as the result of previous trauma. The dog can then be viewed through a lens of empathy and understanding, often lacking for dogs presenting with impulsive, reactive, or aggressive behaviours. Assessment must avoid re-traumatising the animal through exposure to triggering stimuli and, treatment options should include counselling of caregivers on the impact of adverse early experiences, consideration of the window of tolerance, and TIC behavioural modification techniques.

## 1. Introduction

Trauma is a term used to describe a range of experiences that can cause significant distress and have long-lasting negative effects on mental health [1].

Trauma-informed care (TIC) is widely accepted [2,3] as a useful approach in understanding the development of human health and mental health difficulties resulting from adverse childhood experience (ACE) [4]. TIC describes a framework for working with and relating to people presenting with both the physical and emotional signs associated with early trauma, often summarised using the 4 R’s approach: realise, recognise, respond, resist re-traumatisation [5]. TIC differs from previous approaches aimed at modifying harmful behaviours or ameliorating negative emotions because it recognises the importance of considering any underlying trauma experienced by the patient [6]. Instead of asking “what is wrong with you”, TIC considers, “what happened to you” [7]. This approach is a biopsychosocial model integrating research from neurobiology, attachment, trauma, and resilience [8,9]. 

A simplified explanation of this concept considers the “danger-response” of a patient [10]: when a patient experiences actual danger or the anticipation of danger, whether real or imagined, their body responds in a protective manner. This may be recognised as either “hyper- or hypo-arousal” as illustrated in the window of tolerance model [11] where the patient is experiencing autonomic dysregulation because of reminders of their original trauma. The behavioural responses associated with hyper-arousal may be interpreted as agitation, aggression, or sometimes perilous or risky behaviour. In contrast, hypo-arousal results in the patient appearing withdrawn, frozen, or apparently “vacant” behaviours. Both the hypo- and hyper-aroused states are the individual’s attempt to rebalance their autonomic nervous system, returning them to their window of tolerance, where they feel safe, able to think clearly, and make rational decisions. The nature of arousal experienced and demonstrated behaviourally can be situation specific and not necessarily consistent, fluctuating within the same situation and between occasions where the situation ‘appears’ similar. This may result in the patient initially appearing to be animated and aggressive, before seeming to shut down, or vice versa, for example, a seemingly subdued individual suddenly “flipping a switch” and lashing out. 

The imbalance of ascending dopaminergic tracts results in rapid alterations in mood, arousal, and ultimately behavioural responses [12]. A TIC approach would consider that the exaggerated responses seen in an individual patient may be the result of previous trauma. Instead of considering “what is wrong with the individual,” making them behave in a certain way, it focuses on supporting both the patient and those around them, recognising the unpredictable, dynamic, and sometimes difficult to explain, responses to particular triggers and finding practical solutions to minimise the risk posed from future triggering events [5].

This paper considers the possibility that a trauma-informed care approach may be helpful in canine veterinary behavioural medicine; however, this does not exclude the possibility that similar post-traumatic effects occur in other species of animals, with species-specific variation in severity, the cognitive processes involved, and the resultant behavioural signs exhibited.

### 1.1. The Importance of Fear, Anxiety, and Stress as a Welfare Concern in Dogs

Canine behavioural pathologies contribute to both relinquishment [13,14,15,16,17,18] and elective or ‘convenience’ euthanasia in all dogs, but the prevalence in younger dogs is particularly high [19]. This is partially the result of them being less likely to suffer from other pathologies resulting in euthanasia (chronic DJD, neoplasia, etc.), and also the immaturity of the human:dog bond. As such, this constitutes a serious welfare concern for juvenile dogs (those who have not yet reached physical maturity). Caregiver reporting on the spectrum of fearful–aggressive behaviours often describes ‘unpredictable’ or excessive/over the top responses [20] to a situation/animal/person (herein described as the trigger), to which the behaviour is directed. Estimates of the prevalence of fearfulness and aggression vary across different areas and studies estimate it to be between 20 to 50% of the dog population [21,22,23,24].

Both fear and anxiety are unpleasant emotional reactions to the actual presence (fear) or potential presence (anxiety) of a threat [25]. Fear and anxiety are expressed by many of the same physiological and behavioural signs and there is no empirical data on distinguishing between them [26]. It is therefore difficult to determine whether the trigger is the direct source of the dog’s negative reaction or if the dog is associating another, apparently innocuous stimulus (trigger), with the original trauma and reacting as if they are anticipating trauma or danger to the same degree as they have experienced it before. This is an exaggerated form of a normal self-protective or defensive response, which becomes maladaptive for the patient when it prevents them from having normal or healthy interactions with their environment. Some examples might include: a dog who experienced either verbal or physical abuse by a man, and who then mistrusts all men; or a dog who was frightened by the sound of a firework as a puppy, and who will not leave their home or garden in the autumn or winter months, due to the multiple sensory triggers of the season, which they directly associate with their initial fearful exposure to fireworks; equally, one traumatic event associated with travelling to a veterinary setting for treatment, can result in animals being triggered by all future car travel.

### 1.2. Early Experience and Neurobiological Development

Canine behaviour has a significant genetic component and thus, behavioural genotype often correlates closely with the behavioural phenotype exhibited [27,28]. Where there is a genetic predisposition for fearfulness and anxiety, those individuals are at higher risk of experiencing difficulties when faced with potentially stressful or fear-inducing experiences [29]. Environmental factors, such as maternal care, owner experience, habituation, and training also have a significant impact on behavioural phenotype [30,31,32], so where any of these factors are inadequate for the developing puppy or juvenile dog, early experiences can be interpreted adversely. The behavioural phenotype displayed and the severity of any fearful response will be the combined result of that individual’s behavioural genotype, early environment, and epigenetic effects [33].

Trauma can be experienced by an individual at any time in their life. The developing puppy is, however, more vulnerable than an adult dog due to their behavioural immaturity and pliability [34,35] and the absence of resilience [36]. The experiences and behaviour of dogs during their first year of life are crucial in determining their later behaviour and temperament [37]. A maladaptive response to stress during puppyhood, referred to as a toxic stress response [38], plays an important role in the pathway from early adversity to physical and behavioural pathologies. Such toxic stress can have damaging effects on learning, behaviour, physical, and mental health across the lifespan of the animal [39].

### 1.3. Behavioural Resilience

The ability of an organism to adapt to its environment is integral to its survival [40]. Daily life involves confrontation with changing situations that can be physiologically or psychologically challenging. To cope with these ‘physical or perceived threats to homeostasis’ [41] the ability to alter the function and expression of neuronal genes has been developed in the form of molecular and behavioural stress responses [42]. This is advantageous because it allows rapid behavioural, autonomic, and cognitive CNS responses to stressful circumstances, followed by prompt re-establishment of homeostasis and a functional steady state. Both genetics and environmental factors will contribute to the individual’s ability to develop and sustain emotional and behavioural resilience [43].

The offspring of mammals exposed to stress whilst pregnant may experience several physiological consequences, as their bodies are prepared for entering a potentially hostile environment: accelerated growth to compensate for reduced maternal investment [44]; increased risk of psychiatric disorders [45]; and long-lasting changes in the white matter of the hippocampus [46].

While enhancing neuronal communication and promoting survival, this can impact neuronal function and integrity, both immediately and in the long-term. Consequences of stress, such as its negative influences on cognition and emotional stability, are an issue of major relevance in human health and the development of mental resilience, which is a necessity for re-establishment of emotional homeostasis. Chronic and/or severe activation of the stress response early in life has been shown to be potentially injurious in both humans and experimental animals [47,48,49]. 

### 1.4. Adverse Early Experience in Humans

Adverse childhood experiences have been described in human psychiatric medicine since 1995 [50]. Commonly referred to as ACEs, these are potentially traumatic events that occur in utero, childhood, and adolescence, with a spectrum that is extensive but includes physical, emotional, or sexual abuse; witnessing violence; having a family member attempt or die by suicide; and growing up in a household with substance use (or abuse), mental health problems, or instability due to parental separation, divorce, or incarceration [51,52]. In people, multiple ACEs can be experienced by the same individual, over time, or as a result of a single serious event. The long-term impact of such will be influenced by the severity of the ACE, the cumulative effect of multiple ACEs, the support system available to the patient, and their individual genetic and developed resilience.

Stress experienced by a pregnant woman has been shown to have a profound impact on their developing foetus. Prenatal maternal stress and depression can impact the developing foetus, the effects of which can be lifelong and impact future generations [53]. Maternal stress triggers a strong physiologic response that causes the release of cortisol and stress-related neurotransmitters. The release of these neurotransmitters can have a profound and often harmful impact on the growing foetus [54], which has been linked to subsequent stress reactivity and psychopathology later in life [55]. The placenta attempts to filter and protect the foetus from exposure to stress-related effects attributable to these neurotransmitters, but high levels, prolonged exposure, or a compromised placenta limit this protection. It has been suggested that this effect is in fact preparing the foetus for survival in a similarly stressful postnatal environment [56]. 

Several studies have explored the adverse influence of prenatal maternal depression on neurodevelopment in childhood, highlighting increased reactivity [57], more difficult temperament [58], greater negative affectivity [59,60], and high cry reactivity [61]. Furthermore, elevated temperamental domains of activity and emotional reactivity have been linked to depression, anxiety, attention deficit hyperactivity disorder, and conduct disorder later in childhood [62,63,64].

The purpose of ACE screening in human medicine is to identify the likelihood that a patient may be experiencing toxic stress physiology. In and of themselves, the ACEs are not the outcome of interest, it is the long term physiological and neurological effects that are important [65]. 

The trauma-informed care approach includes realisation of how ACEs affect both health and behaviour, encompassing a more holistic approach to medicine; recognition of clinical symptom presentation, particularly when it appears to be difficult to explain via any other aetiology; and adjustment of the health care provider’s approach to treatment to ensure they do not re-traumatise patients when delivering evidence-based care [2]. 

### 1.5. Adverse Early Experience in Dogs

Through comparable mammalian mechanisms to those described in humans, it is plausible that adverse early experiences in dogs that result in toxic stress physiology can have similarly long-term negative impacts on health, behaviour, and resilience. This could include chronic stress, in utero and into puppyhood, multiple “low impact” adverse experiences or a single event, the severity of which is such, that the HPA is activated in a heightened manner and with similar, long-lasting effects. 

As yet, there are few studies on the impact of toxic stress physiology in puppies. Studies in rodents and human children have shown that chronic glucocorticoid excess, including pre- and perinatally, interferes with learning at a cellular level [66,67,68,69], compromising development of the hippocampus and amygdala; therefore, associational learning and fear modulation are altered. This results in memory impairment and compromises fear conditioning [69,70]. Early, pronounced stress has a direct impact on brain pathology and the consequent behavioural changes. 

Adult dogs with naturally occurring hypercortisolism (Cushing’s disease) show disorientation, altered social interactions, disrupted sleep–wake cycles, house-soiling, compulsive behaviours, depressive behaviours, anxiety, and difficulties with memory and learning [71]. Iatrogenic hyperadrenocorticism, from use of steroid medications, results in dogs who are: significantly less playful, more nervous/restless, more fearful/less confident, more aggressive in the presence of food, more prone to barking, more prone to startle, more prone to reacting aggressively when disturbed, and more prone to avoiding people or unusual situations [72]. If these effects are consistently reported in adult dogs with fully developed brains exposed to high levels of blood cortisol, the impact of elevated blood cortisol on the rapidly growing and vulnerable brains of puppies could also be significant and worth further scientific scrutiny.

Puppies born to dams who have experienced significantly high or chronic levels of stress during their gestation period could be at risk, particularly if the stress results in reduced “parenting capacity” which is a risk in both pregnant humans and bitches [73,74]. During the first weeks of life, poor maternal behaviour has an impact on the puppies’ cognitive development [75]. As a result of the crucial nature of both the pre/post-partum care of canine dams, which corresponds with the sensitive neurodevelopmental landmark phases of puppy development, it is vital that puppies and their dams be exposed to humane conditions that minimise the risks of excess stress and fear [76]. As we learn more about epigenetics and the impact of environmental conditions resulting in gene expression, it is easier to understand why puppies raised in “less than ideal” conditions will almost certainly be more vulnerable [77]. Where the environment is stressful for the pregnant/nursing dam, and her puppies (noisy, strong-smelling enclosures, lacking space, lacking comfort, lacking optimal thermal conditions, lacking the optimal physical/mental/social stimulation) presence of any genetic predisposition for fear, anxiety, and stress, will be more likely to be activated [78]. Poor maternal care, where there is mismothering, illness, or absence of the dam, will impact intraspecific socialisation, whether there are appropriately matched littermates present or not [75]. The absence of, limited, or negative human–dog interactions and handling [79], including fear-inducing or painful situations commonly experienced by puppies under 12 weeks of age [80,81], tail docking, dew claw removal, separation from dam/siblings/familiar environment, transport, injections, and implanting of microchips [82] could negatively impact the puppies’ future relationships with people. 

Providing an excellent environment and care of a pregnant dam through pregnancy, nursing of her pups, and then natural weaning, is a time-consuming task requiring skill, experience, and training, to mitigate the inherent risks described above. This may be difficult to achieve, cost-effectively, in larger scale puppy rearing units, with large numbers of breed stock housed in proximity and high animal: caregiver ratios [83].

### 1.6. Adverse Experiences or Traumas in Dogs

The study of adverse events, described as ‘traumatic’ in the canine literature, has focused on a combination of experimentally induced trauma, and naturally occurring traumatic events, such as experiencing natural disasters such as earthquakes [84]. Exposure to experimental traumas was unpredictable and uncontrollable, resulting in greater fear and defensive responses, agitation, tachyarrhythmias, phobias, generalised learning deficiency in relation to escape and active avoidance behaviours, passivity, impaired cognition and learning, and persistent signs of trauma such as tremor, anorexia, intermittent howling, excitement, and recurrent diarrhoea [85,86,87,88,89,90,91,92,93,94,95]. The signs continued over time, without the presence of triggers, demonstrating the long-lasting, chronic influence of the trauma, long after exposure to the traumatic incident. 

Naturally occurring traumatic events and the resultant “post-traumatic stress disorder-like” signs in dogs have been hypothesised about for some considerable time. In 1979, Wagner [96] hypothesised that trauma-trigger responses in animals are intrusion signs, resulting in both an exaggerated and avoidance response to those triggers, which are likely to be expressions of them having re-experienced the original traumatic incident. The evidence to support the occurrence of post-traumatic physical/behavioural pathologies in dogs is, however, a newer development in veterinary behavioural medicine [97,98]. It has been investigated in working dogs utilised in bomb detection or post-disaster deployment [98,99,100] whereas their human-handler counterparts have been diagnosed and treated for PTSD for many years [101]. 

Studies in chimpanzees have shown that adverse early experience impacts on both development of social skills and resilience: orphaned juveniles lacking in the safe and facilitating social environment provided by their mother, were less equipped for fine-tuned social play [102] (assayed by a higher prevalence of aggression) and deciding whom they entrusted to keep their body clean [103] (grooming behaviours). Identification of evidence to support the presence of behavioural clusters similar to PTSD and depression in chimpanzees [104] has contributed to the argument that a diagnosis of complex PTSD in chimpanzees is consistent with descriptions of trauma-induced symptoms as described by the DSM-IV and human trauma research [105] and has led to the emergence of trans-species psychology inclusive of both humans and other species.

### 1.7. Application of a Trauma-Informed Care Approach in Dogs

Application of a trauma-informed care approach might be particularly useful in the assessment and rehabilitation of both shelter dogs and those presented with problematic behaviours (as presented by their caregiver), as companion dogs, or those with a working or assistance dog role.

Many shelter dogs are collected as strays or relinquished by caregivers who can no longer provide that care, for a multitude of reasons. Often the history accompanying these dogs is incomplete or non-existent and this provides additional challenges for those attempting to rationalise the behaviours they are exhibiting. The owners or caregivers of dogs presented for problem behaviours may have equal difficulty understanding why their dogs are reacting to what they (the caregiver) perceive to be a harmless stimulus, or with ‘apparent’ aggression (again, the caregiver’s interpretation).

Dog caregivers presenting their dog for behavioural assessment and therapy may be divided into four categories: They have owned their dog from puppyhood (generally 6–12 weeks of age) and can recall a specific event or series of events that has resulted in their puppy/dog becoming anxious.They have acquired their puppy or dog, beyond 12 weeks of age and are directly aware, or have been informed of a specific event or series of events that may have resulted in their puppy/dog becoming anxious.They have owned their dog from puppyhood (generally 6–12 weeks of age) but are unaware of a specific event or series of events that has resulted in their puppy/dog becoming anxious.They have acquired their puppy or dog beyond 12 weeks of age and are unaware of a specific event or series of events that has resulted in their puppy/dog displaying signs of anxiety or fearful behaviours.

Dogs in the first two categories have a plausible rationale for their anxiety, with or without the addition of early trauma (in utero and during the sensitive phases) to explain the aetiology of the behaviour. Even if the dog caregiver cannot specifically recall a traumatic event in their puppy’s history, it can be determined- with some certainty, by a clinician collecting a medical history exploring aspects of the breeding environment, social isolation, early events, etc. Behaviour exhibited by the last two categories of dog, could potentially be explained by early trauma the dog was exposed to themselves as a puppy, or where the dam had experienced trauma during pregnancy or whilst nursing her puppy/puppies during the sensitive phases. They could equally be the result of trauma experienced by the dog, without their caregiver being aware (either where the caregiver was absent altogether, distracted, or not behaviourally sensitive to the puppy or dog’s response). Irrespective of the category to which the puppy or dog can be categorised here, gestational trauma and early experience could have an influence on the behavioural phenotype of the adult dog. Information about what triggers an individual patient [105] may prove helpful in managing their behaviour. However, because of the cumulative nature of triggering and the individual differences associated with the response to trauma, they may serve as a guide, but not necessarily the solution to the problem. Adoption of a TIC approach for canine behavioural medicine should therefore focus on the management of the case and less on the inciting cause(s). 

In the absence of a corroborative history, we propose that an approach which considers “that there is a likelihood that this puppy/dog has experienced trauma” would be more helpful and productive than “I don’t understand why this dog is behaving this way”. This is the lens of empathy which utilisation of a TIC approach provides.

Equally, when a dog is presented with a known history of trauma, neglect, or abandonment, it could be helpful to formulate strategies which consider the potential current and future impact this might have on their psychological and physical health, thereby setting the patient or animal up for success, rather than waiting for the inevitable signs of behavioural pathology when the dog is placed in a challenging situation.

By applying a TIC approach in assessing these animals, the following scenarios may be considered: the dog may have experienced in utero trauma due to the environment accommodating their dam; or they may have experienced trauma as a puppy, with or without the care provided by their dam or breeding establishment. This could include the effects of mismothering or the absence of positive habituation, in healthy, stimulating, and safe environmental conditions. TIC recognises the long-term impact of fear-inducing experiences in early life, in the absence of behavioural resilience which develops with time, stability, and supportive environmental and social conditions. TIC also acknowledges that the response to trauma is completely individual, based on their unique combination of genotype and early environmental conditions, so that exposure to any type of negative experience as a juvenile animal could result in long-term behavioural pathology.

In veterinary behavioural medicine there are currently several potential differential diagnoses which might be considered in response to trauma: fearful avoidance behaviour [106], fear aggression [107], generalised anxiety disorder [108], panic disorder [109], PTSD [110], PTS-like disorder [99,111], phobias [112], and also some compulsive disorders [113]. Differentiation between these conditions can be very difficult, and various researchers and clinicians may favour one over the other for a variety of reasons. What is consistent between them all is the underlying aetiology, where the patient has experienced trauma, on a spectrum of severity and with various behavioural manifestations.

### 1.8. Modification of Current Treatment Approaches for Anxious/Fearful Dogs to Incorporate TIC

Veterinary behavioural medicine currently uses a combination of medical interventions and behaviour modification techniques for treating fearful animals [114,115,116], which aim to reduce physiological responses to a stressful situation and over time, desensitise the patient to the triggering situation and ideally, ultimately countercondition, so that at worst, exposure to the trigger results in a manageable, neutral response for both the animal and their caregiver, and in the best case scenarios, the negative can be converted into a rewarding situation for that patient, depending on the severity of their baseline state. This approach relies heavily on pet caregiver empathy and compliance [117]. 

Assessing dogs in the care of an animal shelter or rescue organisation will have the additional challenges associated with the absence of social support from a permanent caregiver, balanced against the training, empathy, and capabilities of those working with rescue dogs as a vocation [118]. Nonetheless, would consideration of the 4 R’s approach—realise, recognise, respond, resist re-traumatisation [119]—be helpful in the management of anxious/fearful canine patients? This is not an approach currently documented for use in dogs and this paper will consider whether it is worthy of future consideration and development.

## 2. Exploring Use of Trauma-Informed Care Approaches in Dogs

### 2.1. Adverse Early Experiences (AEE)

Many of the examples of adverse childhood experience can also be experienced by dogs, whether living with an caregiver or as a stray/roaming, independent animal. Due to the differences in cognitive ability and behavioural adaptation, the responses will not be equivalent, but serve as examples or a guide in considering the impact of adverse early experiences (AEEs) in dogs. Table 1 describes the results of a search on adverse early experiences for children and dogs, indicating that few areas have previously been identified or explored in relation to canine welfare. The spectrum of ACE in children is huge and diverse in terms of severity. Instead of listing every category of ACE and whether the equivalent effects have been documented in the canine literature, Table 1 lists six of the main groups of ACE, which have an equivalent consideration in canine behaviour and welfare. All the groupings listed were considered to be highly likely or already supported by the literature, as being relevant to dogs. 

### 2.2. Signs or Symptoms of Trauma in Humans and Dogs

A 2017 study [161] looking at the ordinal risk of a list of potential health outcomes associated with ACE in humans found a moderate risk (OR 2–3) of smoking, heavy alcohol use, poor self-rated health, cancer, heart disease, and respiratory disease, strong associations (OR 3–6) with sexual risk taking, mental ill health, and problematic alcohol usage, and the strongest risks (OR 7+) for problematic drug use and interpersonal and self-directed violence (to include self-harm and suicide). Systematic investigations of the impact of AEE on dog physical health have not been conducted here, but, understanding the correlations between trauma and physical illness is developing within veterinary behavioural medicine [162,163] and worthy of future consideration and research.

The behavioural and mental health impacts of ACE are shown in Table 2, along with the likely applicability to dogs. 

Table 2 highlights that many of the symptoms of trauma demonstrated in humans have also been described in the canine literature as well; however, there are many that were not. This may be explained by the differences in cognitive capacities and the internal processing of trauma that humans are capable of, and dogs are either not known to be capable of, or where they are, it is to a lesser extent. There is however significant scope to explore this further. Consideration of the potential for adverse early experience to cause a range of “danger responses” in dogs, which cannot otherwise be explained rationally by either caregiver or clinician, would be a particular benefit supporting inclusion of TIC in canine behavioural management.

### 2.3. Treatment Strategies Utilising a Trauma-Informed Care (TIC) Approach and Potential Relevance for Canine Welfare

Current behavioural modification techniques for animals with a history of trauma/anxiety/fearful behaviour might include a combination of cognitive behavioural therapy (CBT) and anti-anxiety or mood modifying medications. 

A study evaluating the feasibility, acceptability, safety, tolerability, and potential clinical effectiveness of trauma-focused cognitive behavioural therapy (CBT) in young people [202], reported that 85% of participants “found it helpful.” Many of the TIC-centred treatments described in the human literature would not appear to be suitable for integration in canine behaviour modification approaches, such as talking-based cognitive behavioural therapy, and structured sensory interventions like drawing, imagery, and other forms of expressive art [203]. However, there have been recent attempts to translate eye movement desensitisation and reprocessing (EMDR)—which requires the patient to identify multiple aspects of the traumatic memory and an alternate, desired, positive self-representation [204]—for use in dogs, utilising approaches developed for pre-verbal children [205,206].

Use of the TIC lens, whereby a canine caregiver or treating clinician may be aware of the effects of trauma, recognise its signs, understand how it informs individual responses, and actively try to prevent re-traumatisation are the tenets of trauma-informed care and could be integrated into canine behaviour modification techniques [207,208,209]. In addition, it is plausible that a TIC approach could resonate with owners more than traditional non-TIC approaches, thereby encouraging help-seeking and compliance with behavioural interventions [210]. 

Development of new TIC approaches for use in both shelter dogs and those presented by their caregivers for behavioural therapy, as well as adaptation of current assessment and treatment modalities, to remove the risk of re-traumatisation, would significantly improve canine welfare and potentially caregiver compliance, for both current and new dog caregivers.

## 3. Practical Application of the TIC Approach for Dog Management

Viewing problematic canine behaviours through the lens of early trauma does not excuse or trivialise them, instead it enriches our understanding of how canine behaviour develops, informs intervention strategies, and improves clinical outcomes.

### 3.1. Importance of Pre-Acquisition Advice for Dog Caregivers

Improved acknowledgement and understanding on the importance of the in utero environments and the care of the dam, both pre- and post-partum within the dog breeding, veterinary, and dog owning communities can only be beneficial for the welfare of the dogs in their care. Considering the importance of ‘behavioural health and prophylaxis’ will benefit dog caregivers and help ensure their puppies are better prepared for life in their new domestic environment.

In child psychology, there is an understanding that “it can be too late, but never too early” in safeguarding the mental health of babies and infants [211] (Acquarone, 2004), so too, this could be argued, for the safeguarding of canine mental health. When the mother of a human infant is experiencing acute stress during pregnancy, resulting in the foetus being “incubated in terror [212]”, often the causes of the stress are unavoidable or difficult to remove [213]. When a potential dog caregiver is making the decision about sourcing a puppy to join their family, the time and diligence invested in getting advice and researching breeders or other sources of puppies should be taken without any pressure at all [214].

Avoiding problems associated with traumatic early development, in utero and during the vital sensitive phase between birth and weaning, will help ensure the puppy has had the best possible start in life. This in turn improves outcomes for the breeder in terms of reputation: the new dog caregiver, because their puppy is less likely to have problems and ultimately, the puppy will have improved mental wellbeing and the best possible chance of developing healthy resilience. From there, the puppy’s continued development will rely on the care taken by their new caregiver and those involved directly and indirectly in their care [215,216]. Caution should also be considered about whether puppies who have experienced early trauma, contributing to behavioural issues in later life, will be covered by pet insurance providers [217].

### 3.2. Separation from the Dam, Litter, Care Giver, and Familiar Environment

Opinions vary on the ideal time for weaning, separation of pups from their dams and litter mates, and transport to their second or new home [84,218] (varying in the range of 6–16 weeks and generally recommended as 8–12 weeks). These practices are considered routine and, although there may be some anecdotal advice given to dog caregivers about minimising the associated stress for the puppy, there does not appear to be scientific evidence to inform the advice given. Breed-specific information on variation of timing in the sensitive phases [219] for the huge spectrum of dog sizes, types, and maturities, would help update the current advice given. Transport of puppies from the property they were born on to the home they are intended to stay in varies considerably, in terms of geography, duration, mode of transport, care pre/during/post transport. Information on the optimal age for transport, to minimise the detrimental effects on the puppy, appears to be lacking. There are a number of unknowns here, which could be explored for the advancement of canine welfare.

### 3.3. Procedures Routinely Carried out on Juvenile Dogs

In the UK and Ireland, dogs must be microchipped and registered on an authorised database before 12 weeks of age, or earlier if leaving the property they were born from 6 to 7 weeks onwards [220,221]. Equally, routine vaccination of puppies usually commences between 4 and 12 weeks of age [222] (NOAH, 2023), coinciding with normal puppy sensitive phases. The manner of handling puppies during these procedures, to include catching, lifting, restraint, skin tenting, insertion of the needle, and potential use of distractors [111] can potentially have long lasting effects on the puppy’s acceptance of restraint or any invasive procedures (e.g., annual booster vaccinations) and yet, this has traditionally been considered as routine and low risk and therefore not necessarily given the time, care, and investment of expertise and training that it deserves.

If potentially more invasive procedures are to be carried out, such as tail docking, dew claw removal, ear cropping, etc., depending on the breeds involved and the local legislation in place, even more attention should be given to use of appropriately trained personnel, local anaesthesia, sedation of the puppies, and adequate use of analgesia, to minimise the risks of both physical and mental/emotional damage to the puppies involved.

### 3.4. Assessment of Dogs in Animal Shelters

Various means have been developed of assessing dogs in animal rescue shelters/dog pounds, to triage animals for either euthanasia or potential rehoming [223,224,225]. Utilising a TIC approach would ensure that testing considers, and ideally avoids, the potential detrimental effects of exposure to triggers which could incite a “danger response” in the animal, resulting not only in acute stress for the dog but also an increased likelihood of the outcome or decision being unfavourable for the dog too.

Consideration for the fact that shelter dogs have, as a minimum, lost their previous caregiver, and in many cases experienced multiple other sources of trauma, means that basic empathy should be applied to every stage of their care. On arrival, they will require time to adjust to their new environment, diet, care givers, and proximity to other dogs or species of animals. Ideally their care should be kept stable, minimising the risk of triggering from novel stressors (new kennels, new handlers/trainers/care givers, and to other potential changes to their routine).

Assessments of “temperament” should be delayed until the dog has had the opportunity to recover from their previous trauma (e.g., living as strays, capture and transport to the shelter or foster facility, separation from their caregiver) before any “meaningful” behavioural assessment can be made, particularly when that decision could potentially be one of life or death (euthanasia).

Incorporating TIC into the assessment approach should pay particular attention to the euthanasia decision-making process. Where dogs exhibit signs consistent with previous trauma, they should universally be afforded appropriate empathy. Some will be suitable for, and responsive to rehabilitation, others will not. Where a dog displays significant stress when exposed to triggers, which cannot be avoided and either medical management is not feasible, or has not given the desired response, the option for euthanasing the dog, to avoid future suffering, should be available.

Instead of exposing a shelter dog to a sequential list of potentially triggering objects or triggers, to assess how they react, they should all be considered to be “potentially traumatised” and triggering experiences avoided altogether, to protect the welfare of the animal. Triggers may arise naturally during routine care of the dog, for example, reluctance to have their feet touched; fear of men with beards; white vans, similar to the one they were transported to the shelter in, and so on. Identifying these triggers and documenting them on the dog’s behavioural profile, will be valuable information in selecting suitable new caregivers and also for those new caregivers, in both avoidance and desensitisation to those triggers, where avoidance is either impossible or compromising for the welfare of the dog.

If the acts of relinquishment, abandonment, and loss of their caregiver result in a traumatic experience for dogs in shelters, this in combination with any of the other adverse events experienced by rescue dogs means that they should universally be treated with the appropriate TIC approaches and empathy, by both the rescue organisations and those becoming their new caregivers. This empathy and TIC approach might also be applied by companies providing pet insurance, so that despite a traumatic history, they are still eligible for all of the benefits afforded by being fully covered by pet insurance [226].

### 3.5. Procedures Carried out on Shelter Dogs

Many shelter dogs will require multiple, potentially stressful, procedures to happen as part of their care in the shelter and in preparation for rehoming. This might include vaccination, worming, microchipping, neutering, and grooming to various degrees of difficulty and potential discomfort, etc. These tend to occur right away and in quick succession, so that the dog is “ready to go” as soon as an alternative home has been secured. A TIC approach could be applied at various levels in the management of these procedures. Initially, nothing potentially scary or painful should be performed on the dog until they have had time to recover from their entry to the shelter. Except for acute veterinary pathological conditions, everything else can wai,t and be timed with sensitivity and planning to minimise the negative impact on the dog.

If there is any indication at all that a dog may be euthanased rather than rehabilitated, they should not be challenged unnecessarily. Additional stress from handling, restraint, and potentially painful or scary procedures increases the dog’s physiological stress and increases the risk of them “failing” an assessment of temperament. If microchipping, neutering, etc., can wait, then it should. This will improve the short-term welfare of the dog, improve their chances of “settling in” so they perform well if or when testing occurs, and also, saves the organisation money for procedures or medications “lost” on a dog who is then euthanased.

If sedation or a general anaesthetic is required for neutering, dental work, lump removals, or orthopaedic procedures, as many as possible of the “other” necessary procedures should be carried out while the dog is asleep and unaware (e.g., microchipping, grooming etc.). This reduces the stress for the dogs, caretakers, and veterinary teams.

### 3.6. Dogs Exhibiting Problematic Behaviours

Incorporating a TIC approach into the treatment protocols of dogs presented with problematic behaviours, whether owned or in a shelter, would be a novel concept which builds on current practices in veterinary behavioural medicine. These generally combine the following: pharmaceutical management of the physiological stress response, both chronic, as in anxiety or acute, in response to exposure to triggers and; behaviour modification techniques. The TIC-focused strategies would focus on treating the animal for underlying anxiety and any unexplained or “over the top” responses to triggers, irrespective of whether there is a plausible explanation for their behaviour, or not.

Current, potential, or new dog caregivers should have TIC and the influence of AEEs explained to them, potentially using visual aids such as the “window of tolerance” [11] so that they understand: why their dog is behaving as it is; how medication may help widen the window of tolerance; how reframing exposure to a particular trigger, through desensitisation and counter-conditioning can help them reduce and/or redirect their stress response and; how relaxation exercises, EMDR, and other stress-reduction techniques can alleviate an acute stress response, improve dog caregiver empowerment, and strengthen the human–dog bond.

In the same manner as the shelter dogs, a TIC approach would consider whether avoidance of “triggering situations” is possible at all, and if not, is it beneficial or detrimental to the dog’s welfare? Forcing exposure in an attempt to desensitise and counter condition needs to be planned carefully, sensitively, and slowly to ensure both dog and caregiver(s) are coping with the treatment plan. This may result in delays until underlying medical issues are resolved, pain management has stabilised, and if anti-anxiety medications are utilised, that therapeutically effective blood titres and the desired anti-anxiety effects are reached, before challenging the dog.

## 4. Limitations of the TIC Approach

There are two main limitations to the TIC approach: firstly, differentiating stress from trauma and secondly, differentiating traumatised versus non-traumatised canine patients. These are the same arguments that are explored in the human literature, where definitions of stress, crisis, and trauma overlap [227]. Children who have experienced ACEs are less able to discriminate between threatening and non-threatening stimuli in later life, through impaired threat discrimination and deficient reward processing [228].

Individual people and/or animals do not respond to stress in the same way and their response will be determined by numerous factors: personality, temperament, exposure to concurrent stressors, protective factors, adaptability to change, presence of suitable support systems, as well as the intensity and duration of the stressor [229]. Early adverse experience will however have a long-term impact on the development of a number of these variable factors in response to stress, which is any stimulus, internal state, situation, or event with an observable individual reaction, usually in the form of positively or negatively adapting to a new or different situation in one’s environment [230].

If we accept that an individual’s response to a particular stressor or trigger is unique, and determined by a range of factors, almost of all of which are outside of the individual’s control, then differentiating between stress and a life-threatening trauma becomes less important. If the individual perceives a situation to be dangerous or a predictor of a dangerous situation to come, it is no less real for that individual, than a truly life-threatening situation or event. Empathetic management of the individual is the same, whether the threat is real or imagined and as such, use of a TIC approach is still valid.

Differentiating between traumatised and non-traumatised individuals becomes important when “over diagnosis” masks or prevents treatment of a concurrent aetiology or pathological state, therefore compromising the welfare of the individual overall. The risk of this can be minimised by ensuring that physical parameters are tested either before or during psychological assessment to ensure that any underlying medical conditions, particularly those associated with pain, are identified, or ruled out. This provides further evidence for the need to consider veterinary behavioural patients holistically, using a “whole animal approach” and ensuring that underlying veterinary pathology is addressed first, to maximise the potential for behavioural management to prove helpful for the patient.

Equally, one might ask whether the differentiation between traumatised or non-traumatised individuals is the important question, or rather, should it be differentiating between dogs who appear able to “cope with” their environment (and the triggers contained within it) versus those who “appear to struggle to cope with” their environment [231]. Is the previous trauma the important factor, or the ability to cope with stress? If we accept that the combination of genetics, experience, epigenetics, developed resilience, and physiological state at the time of exposure, all influence a dog’s ability to cope with stress, use of a TIC approach, with recognition, response and avoidance of re-traumatising still has scope to improve the situation for the dog, irrespective of their behavioural history.

Similarly, it is believed that many veterinary behavioural patients experience chronic pain, to varying degrees of severity. This is often overlooked, because it is difficult to conduct a meaningful pain assessment on an overtly stressed animal and use of sedation or anaesthesia limits the ability to assess the animal’s response to palpation or manipulation of painful areas of the body [163]. Mills et al. (2020) concluded that “in general, it is better for veterinarians to treat suspected pain first, rather than considering its significance only when the animal does not respond to behaviour therapy”. If it is accepted that trauma is diverse its source, its severity, and the resultant effects on the individual experiencing it, then it could similarly be argued that treating a veterinary behavioural patient utilising a TIC approach routinely would be more beneficial for more dogs, than the few cases where it can categorically be proved that the individual has never been exposed to trauma at all.

## 5. Conclusions and Animal Welfare Implications

Use of a trauma-informed care approach in the management of dogs in rescue shelter scenarios or presented for veterinary behavioural treatment, as either companion or working dogs, is a concept worthy of consideration and more investigation. The benefits of coaching new or potential dog caregivers on the importance of minimising early adverse experiences in puppies would help support the need for responsible breeding practices and careful management of both the sale and transport of young dogs. The management of stray or relinquished dogs would benefit from implementation of TIC to reduce the risk of re-traumatisation and aid in the counselling of both relinquishing dog caregivers, as well as those hoping to adopt a rescue dog, to better understand why dogs sometimes do the things that they do. Over-utilisation of the TIC approach must be a consideration, but where comprehensive physical veterinary diagnostics are used, and the behavioural assessment is carried out by a suitably trained and experienced individual, the risks posed by utilising TIC can be minimised.

## Figures and Tables

**Table 1 animals-14-00459-t001:** Types of adverse childhood experiences with high likelihood of applicability to dogs.

Type of Early Adverse Experiences in Children (REFs)	Equivalent Adverse Early Experience in Dogs	References Identified in the Canine Literature
Deliberate psychological harm	Verbal abuse, neglect	Dogs react fearfully towards human shouting [120]; Effects of fictitious illness on animals, Munchausen syndrome by proxy [121]; Exploring the complexities of animal hoarding and children who abuse animals [122]; Understanding the psychological outcomes of neglect, abuse, and harsh environments in dogs [123]; Postmortem evidence of neglect [124]; The nature of, and criminal justice response to, shooting, beating, stabbing, or throwing of dogs and cats [125]; Determining neglect by assessing bone-marrow fat reserves [126].
Physical harm (Non-accidental injury or pain, sexual abuse)	Physical harm (Non-accidental injury or pain, sexual abuse)	Considers factors raising suspicion of non-accidental injury in animals: vaginal and anorectal injuries and damage to genitalia from sexual abuse [127]; Fracture characteristics distinguishing accidental and non-accidental injury in dogs [128]; Postmortem evidence of non-accidental injuries (blunt-force or sharp-force trauma, gunshots, burns, drowning, asphyxiation, or suspicious intoxications) [129]; Traumatic medical procedures and stressful visits to the veterinary clinic [130]; Considering the evidence of animal cruelty in dogs presented both dead and alive [131]; Collaboration between forensic anthropology and veterinary pathology to investigate non-accidental injury in dogs [132].
	Accidental injuries/accidents	Dogs experiencing the trauma of home fires [133]; Dogs injured during car crash incidents [134]; Acts of nature: wind and storms, rain, and flooding [135].
Family psychopathology	Challenges within the home environment (mental health challenges, substance abuse, inter-personal violence)	‘Spousal violence’ and implications for animal cruelty in the same household [136]; A survey of ‘battered’ women in shelters identifying links with cruelty to dogs [137]; Comorbidity between pet abuse and family violence [138]; Domestic violence and animal cruelty in New Providence [139]; Incidents of animal abuse and neglect in dogs belonging to domestic violence victims [140]; Challenges of owning a dog for psychiatric patients resulting in delays in dogs being treated [141]; Dog ownership and survival following cardiovascular events, where owners’ compromised health results in degrees of neglect and instability for the dog [142]; Dog owners are more likely to delay medical procedures, depending on the availability of social support [143]; Strong dog–owner relationship associated with poorer mental health, anxiety, and depression [144]; Difficulties forming a bond for veterans with their services dogs with comorbid substance abuse and PTSD [145]; Disadvantages and barriers of dog ownership for owners of assistance dogs [146]; Dog ownership can hinder access to services for owners with substance abuse [147].
Unintentional caregiver absence	Owner or caregiver absence due to the following: hospital stays, imprisonment, owner death.	Response of guide dogs to separation from their owner [148]; Examining how imprisoned dog owners relate to their pets [149]; Anxiety in dogs caused by owner absence [150]; Dog owners imprisoned [151]; Effects of death of dog owners on their dogs following cardiovascular incidents [152].
Intentional removal from caregiver	Relinquishment, abandonment or change of owner in dogs.	Risk factors for relinquishment of dogs in the US [153]; Under-developed human: dog bonds resulting in relinquishment [18]; Dogs in UK shelters are more likely to have reported behavioural problems [154]; Factors resulting relinquishment of dogs in Portugal [155]; Factors resulting in relinquishment of dogs in Canada [156].
Bullying or peer violence (threat of or actual physical violence)	Intraspecific, inter-dog aggression	Rehabilitating shelter dogs with reported inter-dog aggression [157]; Risk factors for inter-dog aggression in dogs in the UK [158]; Understanding fear and aggression in dogs [159]; Physiological stress, coping, and anxiety in greyhounds displaying inter-dog aggression [160].

**Table 2 animals-14-00459-t002:** Potential application of human signs of trauma to dogs.

Likelihood of Application of Human Signs to Dogs	Mental or Emotional Symptoms or Signs of Trauma Identified in Humans	Relevance to Canine Behavioural Medicine
Likely to apply to dogs	Aggressive behaviours	Effect of owners challenging already anxious dogs may worsen their tendency to bite [164]; Where a dog has been attacked by another dog in earlier life, they are more likely to mount a vigorous and fast attack [165]; Considering canine personality and previous trauma and their impact on fearfulness and aggression in dogs [166]; Effects of early experience on human-directed aggression in dogs [167].
	Anxiety	Separation anxiety behaviours in dogs [168]; The stress of living with fear and anxiety has negative effects on health and life-span in dogs [169]; Multifactorial nature of fear and anxiety in dogs to include learned experiences during sensitive periods [170].
	Attention deficit/hyperactivity disorder	Age and training experience of dogs showed significant effect when ADHD and non-ADHD dogs are compared [171]; Early social and physical parameters influence the development of ADHD in dogs [172]; Demographic, environmental, and behavioural factors contributing to aggression, ADHD-like behaviour, and repetitive behaviours in Finnish dogs [173]; Serum serotonin and dopamine are lower in dogs with behavioural signs consistent with ADHD-like disorders [174].
	Avoidance behaviours [110] Trauma-related thoughts or feelings; Trauma-related external reminders; Avoidance of triggers; Psychological numbing such as disassociation.	Intensity of heart rate responses and the occurrence of avoidance behaviours in dogs [175]; Escape, avoidance, and learned helplessness behaviour in dogs [176]; Early experience and later avoidance behaviours in dogs [177]; Breed differences in the onset of fear-related avoidance behaviours in dogs [178].
	Depression	Exploring anhedonia in kennelled dogs [179]; Greater periods of waking inactivity of dogs in their home environment [180]; Anhedonia: effect of stress on eating behaviour in domestic dogs [181].
	Memories: recurrent, involuntary, and intrusive memories Traumatic nightmares; Dissociative reactions, flashbacks; Intense and prolonged distress; Marked physiological reactivity after exposure to trauma-related triggers.	Emotional distress resulting in calm behavioural responses but higher cardiac activation in guide dog [182]; Fearful behaviours exhibited by dogs with memory of previous fear inducing/negative experiences [148]; Dogs subjected to neglect, abuse, or severe under-socialisation may develop moderate–severe fearfulness [183].
	PTSD or PTSD-like behaviours	PTSD has been demonstrated in dogs following both experimental and natural trauma [110]; PTSD demonstrated in both household pets and military canines [184].
	Reactivity [110]: Irritable or aggressive behaviour; Self-destructive or reckless behaviour; Hyper-vigilance; Exaggerated startle response; Problems concentrating; Sleep disturbance.	Temperament influences reactivity in dogs [185]; Stimulus reactivity in dogs presented with behavioural problems [186]; Reactivity of puppies towards tolerance of new situations [187]; Effect of owner presence on reactive behaviours of dogs during a stressful experience [188]; Cortisol levels in hair reflect dog’s chronic state of emotional reactivity [189]; Fear and reactivity in working dogs remains consistent between 6 and 12 months of age [190]; Dogs’ emotional reactivity and the dog–owner relationship modulate each other [191].
	Reduced parenting capacity	Influences on maternal care on stress in dogs and rodents [192]; The quality of maternal care contributes to the adaptation of puppies to environmental stimuli [21]; The level of interactions with the dam influences the physiological, cognitive, and behavioural development of their puppies [193]; The peripartum period is stressful, so minimising stress and improving reassurance of the bitch improves her mothering ability [75].
	Rule-breaking behaviours	Interactions between demographics, problematic behaviours, and trainability in dogs [32]; Owner perceptions of guilt as a resultant of canine misbehaving [194]; Problem-solving capabilities in dogs diagnosed with anxiety disorders [195]; Dog-directed parenting skills predicting the resultant behaviour of their dogs [196].
	Stress	Early experience modulates stress in German Shepherds [197]; Effect of maternal care, attachment, and socialisation on responses to stress in dogs [198]; Assessing stress caused by environmental changes in laboratory dogs [199]; Stress responses in dogs exposed to repeated noxious stimuli [200].
Likely but not documented	Distressing thoughts, feelings, or sensations	Empathic-like responding of dogs to distress in their owners [201].
	Difficulty forming emotional attachments	
	Low morale	
	Mistrusting	
	Psychological distress	
Unlikely to apply to dogs	Substance abuse disorders	

## Data Availability

No data collection or storage was involved in this study.

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
