# Peer review of "Potential Benefits of a ‘Trauma-Informed Care’ Approach to Improve the Assessment and Management of Dogs Presented with Anxiety Disorders"

_animals, 2024, doi:10.3390/ani14030459_

Round 1
Reviewer 1 Report
Comments and Suggestions for Authors
I read the paper ‘Potential benefits of a ‘Trauma Informed Care’ approach to improve the assessment and management of dogs presented with anxiety disorders’ with great interest. It is a fascinating topic, and I think there is so much room for improvement in our current management of behavioural issues in non-human individuals. The authors did a great job in describing child/puppyhood trauma, how it can be caused and the effects it can have later in life, and the factors associated with it, and overall, it is a good paper. However, I do have a few questions/suggestions for improvement of the paper, detailed below.
It would be beneficial if the authors add a small section detailing what a TIC approach looks like. What does it mean in practical terms if a person applies a TIC approach to a subject (either dog or human)?
After the introduction and quite detailed explanation of TIC and the comparisons between dogs and people, I expected a discussion to follow in section 3, detailing practical application of TIC to canine patients. The title even suggests that is going to happen in that section. However, that section mostly deals with prevention of trauma in pups, not with dealing with (sub)adult dogs that already have behavioural issues that stem from puppyhood trauma, and how to treat these.
I found it a little disappointing that there is only a very small section (2.3) with some discussion of practical methods dealing with trauma in dogs, and I expected a longer discussion with practical application of methods applied to dogs. I understand that perhaps that information simply does not exist in the scientific literature yet, but it might help readers like me if the authors clearly state that in that section if that is the case, so we are aware that is the extend of the discussion on the matter at this point in time. Maybe the authors could speculate on other methods that could be applied to dogs, even though there might not be evidence of the use of those or their effectiveness. It would then also help to rename section 3, as currently the title creates the expectation that it is a discussion of practical application of trauma management in dogs, not mostly the prevention of trauma in pups.
Page 7, lines 332-334. The 4 R’s response. I would have found it useful to have read about this in an earlier general discussion of management of trauma, rather than on-the-spot there when talking about specifically management in dogs. If discussed earlier, the reader is already aware of it, and the 4 R’s can then be referred to when talking specifically about dogs. I suspect this relates to my earlier comment about adding a section of discussion on what a TIC approach looks like in practice.
Reviewer 2 Report
Comments and Suggestions for Authors
This is still too long and detailed. Trauma Informed Care(TIC), links the type of Care to the type of Trauma. The publication needs more focus on the CARE interventions and how to approach and solve the problem ( how significant is this diagnosis for animal welfare and rehoming dogs. Is TIC curable? Is TIC preventable? Is it hopeless? Should these dogs be euthanazed?) If the research question is: does TIC exist in dogs? The answer is “probably” and “it should be considered”.
Moderate editing of English language required
Author Response
We would be keen to understand which sections you feel are surplus to requirement, so that we can make the paper more concise.
Section 3 is the "speculative" section and it has been expanded slightly to consider the euthanasia decision making processes and also how TIC can be incorporated into treatment approaches- though this is something that may follow, if there is sufficient interest in this concept in canine behavioural medicine.
Changes are highlighted in red
Consideration for the fact that shelter dogs have, as a minimum lost their previous owner, and in many cases experienced multiple other sources of trauma, means that basic empathy should be applied to every stage of their care. On arrival, they will require time to adjust to their new environment, diet, care givers and proximity to other dogs or species of animals. Ideally their care should be kept stable, minimising the risk of triggering from novel stressors (new kennels, new handlers/ trainers/ care givers and to other potential changes to their routine.
Assessments of “temperament” should be delayed until the dog has had the opportunity to recover from their previous trauma (etc. living as strays, capture and transport to the shelter or foster facility, separation from their owner) before any “meaningful” behavioural assessment can be made. Particularly when that decision could potentially be one of life or death (euthanasia).
Incorporating TIC into the assessment approach should pay particular attention to the euthanasia decision making process. Where dogs exhibit signs consistent with previous trauma, they should universally be afforded appropriate empathy. Some will be suitable for, and responsive to rehabilitation, others will not. Where a dog displays significant stress when exposed to triggers, which cannot be avoided and either medical management is not feasible or has not given the desired response, the option for euthanasing the dog, to avoid future suffering, should be available.
Instead of exposing a shelter dog to a sequential list of potentially triggering objects or triggers, to assess how they react, they should all be considered to be “potentially traumatised” and triggering experiences avoided altogether, to protect the welfare of the animal. Triggers may arise naturally during routine care of the dog, for example, reluctance to have their feet touched; fear of men with beards; white vans, similar to the one they were transported to the shelter in, and so on. Identifying these and documenting them on the dog’s behavioural profile, will be valuable information in selecting suitable new owners and also for those new owners in both avoidance and desensitisation to those triggers, where avoidance is either impossible or compromising for the welfare of the dog.
3.5 Procedures carried on shelter dogs
Many shelter dogs will require multiple, potentially stressful, procedures to happen as part of their care in the shelter and in preparation for rehoming. This might include vaccination, worming, microchipping, neutering, grooming to various degrees of difficulty and potential discomfort etc. These tend to occur right away and in quick succession, so that the dog is “ready to go” as soon as an alternative home has been secured. A TIC approach could be applied at various levels in the management of these procedures. Initially, nothing potentially scary or painful should be done to the dog until they have had time to recover from their entry to the shelter. Except for acute veterinary pathological conditions- everything else can wait and be timed with sensitivity and planning to minimise the negative impact on the dog.
If there is any indication at all that a dog may be euthanased rather than rehabilitated, they should not be challenged unnecessarily. Additional stress from handling, restraint and potentially painful or scary procedures, increases the dog’s physiological stress and increases the risk of them “failing” an assessment of temperament. If microchipping, neutering etc can wait, then it should. This will improve the short- term welfare of the dog, improve their chances of “settling in” so they perform well if or when testing occurs and also, saves the organisation money for procedures or medications “lost” on a dog who is then euthanased.
If sedation or a general anaesthetic is required for neutering, dental work, lump removals or orthopaedic procedures, as many as possible of the “other” necessary procedures should be carried out while the dog is asleep and unaware. This reduces the stress for the dogs, caretakers and veterinary teams.
3.6 Dogs presenting with problem behaviours
...
In the same manner as the shelter dogs, a TIC approach would consider whether avoidance of “triggering situations” is possible at all and if not, is it beneficial or detri-mental to the dog’s welfare to do? Forcing exposure in an attempt to desensitise and counter condition, needs to be planned carefully, sensitively and slowly to ensure both dog and owner(s) are coping with the treatment plan. This may result in delays until underlying medical issues are resolved, pain management has stabilised and if anti- anxiety medications are utilised, that therapeutically effective blood titres and the de-sired anti- anxiety effects are reached, before challenging the dog.
Reviewer 3 Report
Comments and Suggestions for Authors
This is an interesting and deep review of the literature on trauma studies in dogs. The authors draw equivalences in trauma and ACE’s in humans to companion dogs. They suggest Trauma Informed Care with less of a focus on the specific trauma inducing event, to instead focus on empathy and understanding in response to behaviors. This review would be useful foundation to further research.
They have missed the literature on PTSD in chimpanzees (eg Van Leeuwen et al 2022, Bradshaw et al 2008, Ferdowsian et al 2011). This might strengthen their argument is it would provide additional evidence for PTSD in nonhuman animals.
Small corrections in typos.
Line 52 individual’s need apostrophe
Line 185 extra period before “compromising”
Line 183-189 - ml check the references.
Table 1 title has extra period
Author Response
Thank you for the proof reading edits- hopefully they have now been amended.
I have added in the chimpanzee references relating to PTSD- thank you!
Reviewer 4 Report
Comments and Suggestions for Authors
The wording of the title already is misleading: "Care", at least for me as a non-native speaker, means treatment, or at least lessening the burden of a patient.
However, the article mostly is a review of existing literature from mostly human studies.
this is a very important contribution to make veterinarians, dog therapists etc aware of what is "out there" in human psychology/psychiatry etc.
However I do not detect in-depth attempts to treat these animals. For the same reason, "management" in the title also should be avoided.
Similarly, "assessment" to me means diagnosis, and again I do not find serious attempts there
It would be helpful for the readers if the authors would describe symptoms (behaviour, physiology...) of these different reasons for trauma, from human studies.
Ideally, these symptoms could, perhaps in a second paper of a series, be used to formulate hypotheses that can be tested in dogs
In a third paper then suggestions for treatmnent, based on part # 2 and derived again from human studies, could be made.
Apart from that, the concept of stress should be covered in more depth ( eg see definitions by Donald Broom)
And, especially under the attempt of asking "What happened..." instead of "What is wrong..", a life history perspective, e g see Taborsky et al 2021, should be introduced. Many of these "traumata" are just adaptations to a different environment, and not symptoms of a negative development.
And: in the list of references some are incomplete, and I also recommend that each article is awarded only ONE number, not a new one each time it is cited again.
Nevertheless, as a literature review it is definitely worth publishing.
Author Response
The sections of incorporation of TIC into the assessment and management of shelter/ problem behaviour dogs has been expanded slightly to address your comments- this was the speculative section and so we were cautious about delving too deeply into that. Hopefully further investigation into the practical application of TIC in these two areas of canine behavioural medicine will follow.
The section on the limitations of this approach has been qualified with an additional paragraph, to include Prof. Broom's work:
Equally, one might ask whether the differentiation between traumatised or non- traumatised individuals is the important question, or rather, should it be differentiating between dogs who appear able to “cope with” their environment (and the triggers contained within it) versus those who “appear to struggle to cope with” their environment [236]. Is the previous trauma the important factor, or the ability to cope with stress? If we accept that the combination of genetics, experience, epigenetics, developed resilience and physiological state at the time of exposure, all influence dogs ability to cope with stress, use of a TIC approach, with recognition, response and avoidance of re- traumatising still has scope to improve the situation for the dog, irrespective of their behavioural history.
Round 2
Reviewer 1 Report
Comments and Suggestions for Authors
The authors did a great job in adding descriptions where necessary to address the concerns I had when I read the paper the first time around, and for me, this added a lot of clarity to the paper. I do not have any major concerns anymore, only a few point where a little more clarification might be beneficial – see below.
Lines 90-91. Do canine behavioural pathologies not contribute to relinquishment and euthanasia of already mature dogs as much?
Lines 172-173. If the placentra filters and protects the foetus from the effects of the neurotransmitters, than how can they reach the foetus to have a harmfull impact?
Lines 587-580. If it is accepted ….. trauma at all’. Should that not read: ‘…. can be proved that the individual has been exposed to ….’, rather than ‘has never been exposed to’.
Author Response
1.Canine behavioural pathologies contribute to both relinquishment [13,14,15,16,17,18] and elective or ‘convenience’ euthanasia in all dogs, but with a higher prevalence in younger dogs [19], who are less likely to suffer from the other pathologies resulting in euthanasia (chronic DJD, neoplasia etc.) and as such, this constitutes a serious welfare concern for juvenile dogs (those who have not yet reached physical maturity).
2. The placenta attempts to filter and protect the foetus from exposure to stress-related effects attributable to these neurotransmitters, but high levels, prolonged exposure or the placenta is compromised, this protection is limited.
3.If it is accepted that trauma is diverse its source, its severity and the resultant effects on the individual experiencing it, then it could similarly be argued, that treating veterinary behavioural patients utilising a TIC approach routinely, would be more beneficial for more dogs, than the few cases where it can categorically be proved that the individual has never been exposed to trauma at all.
I think this is correct- firstly it's almost impossible to say that a dog has never been exposed to anything potentially traumatic or triggering- unless reared in a lab or controlled environment, with constant supervision. Secondly, if we treat them all, just in case, for the few it does not apply to , their welfare is still enhanced by sensitive and empathic care.
I hope that makes sense?
Reviewer 4 Report
Comments and Suggestions for Authors
The paper has improved, in my opinion significantly.
I wouzld still be happier with another title, because I still do not find a lot of info on diagnosis or treatment in the text. But maybe a native speaking editor can better judge this, regarding the wording.
And I would be happier if the term "owner", "owned" etc could be avoided. A sentient being should not be the property of someone else. Caregiver, companion etc would be better suited
Author Response
I have considered the comments from the initial review as "permission" to explore more on how assessment and care can utilise TIC. I was cautious initially because i cannot evidence it- but we have to start somewhere?
I will attempt to swap caregiver from owner- I think I have used both throughout, but I understand your sentiment in this.